# DL4-μbeads induce T cell lineage differentiation from stem cells in a stromal cell-free system

Ashton C. Trotman-Grant[1,2], Mahmood Mohtashami[2], Joshua De Sousa Casal [1,2], Elisa C. Martinez[2], Dylan Lee[2], Sintia Teichman[1,2], Patrick M. Brauer [2], Jianxun Han[2], Michele K. Anderson [1,2] & Juan Carlos Zúñiga-Pflücker [1,2✉]

T cells are pivotal effectors of the immune system and can be harnessed as therapeutics for regenerative medicine and cancer immunotherapy. An unmet challenge in the field is the development of a clinically relevant system that is readily scalable to generate large numbers of T-lineage cells from hematopoietic stem/progenitor cells (HSPCs). Here, we report a stromal cell-free, microbead-based approach that supports the efficient in vitro development of both human progenitor T (proT) cells and T-lineage cells from CD34[+] cells sourced from cord blood, GCSF-mobilized peripheral blood, and pluripotent stem cells (PSCs). DL4-μbeads, along with lymphopoietic cytokines, induce an ordered sequence of differentiation from CD34[+] cells to CD34[+]CD7[+]CD5[+] proT cells to CD3[+]αβ T cells. Single-cell RNA sequencing of human PSC-derived proT cells reveals a transcriptional profile similar to the earliest thymocytes found in the embryonic and fetal thymus. Furthermore, the adoptive transfer of CD34[+]CD7[+] proT cells into immunodeficient mice demonstrates efficient thymic engraftment and functional maturation of peripheral T cells. DL4-μbeads provide a simple and robust platform to both study human T cell development and facilitate the development of engineered T cell therapies from renewable sources.

[1] Department of Immunology, University of Toronto, Toronto, Ontario, Canada. [2] Sunnybrook Research Institute, Toronto, Ontario, Canada.
✉email: jczp@sri.utoronto.ca

T cells are an important cellular component of the immune system that can be harnessed for both regenerative and immunotherapeutic approaches[1–3]. Currently, T cell-based immunotherapies generally rely on autologous T cells, limiting their use[4,5]. To overcome this challenge, technologies for the in vitro generation of T-lineage cells from stem cells have been developed[6–10]. However, there remains a need for the scalable production of T cells in a non-xenogeneic stromal cell-free system that is compatible with clinical applications.

T cells develop from bone marrow-derived progenitors that seed the thymus. The thymus provides a specialized micro-environment wherein thymic epithelial cells (TECs), expressing Delta-like 4 (DLL4), interact with Notch-1 receptor-bearing progenitors to induce T cell commitment and differentiation[11]. Several stromal-cell based approaches mimic the thymic environment by providing Notch ligands, including the OP9-DL4 co-culture system[7,12] and the recently developed artificial thymic organoid system[8,13]. These approaches enable Notch signaling to initiate the T cell program in hematopoietic stem/progenitor cells (HSPCs) in the absence of the thymus. However, the use of serum-containing medium and xenogeneic stromal cells in these systems limits their use for clinical translation.

The development of stromal-cell free systems similarly recapitulates the requirement for Notch signaling, such as the use of plate-bound Notch ligands[7,14–18] and adhesion molecules, like VCAM1[19], for the generation of T-lineage cells and progenitor T cells (proT), with proT cells being capable of thymus engraftment in immunodeficient mice. Importantly, plate-bound approaches are limited by the surface area of tissue-culture plates, which presents challenges for large-scale production[20,21].

An ideal alternative would be a technology capable of generating T-lineage cells from multiple sources of stem cells using a simple serum- and stromal cell-free approach that is amenable to large-scale suspension cultures. Here, we report the development of a bead-based in vitro system that supports robust differentiation of mouse and human HSPCs and human induced pluripotent stem cells (iPSC) to yield proT and T-lineage cells. Therefore, the DL4-µbead culture system provides an avenue for the clinically relevant generation of T cells from engineered PSC lines.

## Results

**DL4-µbeads induce Notch signaling and T cell development from mouse HSPCs.** Previous stromal cell-free strategies have relied on two-dimensional (2D) tissue culture platforms using DLL4 Fc-fusion proteins (DL4-Fc) immobilized on tissue-culture plates[7,9,18,19]. We evaluated whether DL4-Fc immobilized onto the surface of microbeads could induce Notch activation. The DL4-Fc construct was designed with a BirA recognition sequence (AviTag) at its C-terminus, enabling enzymatic conjugation with a single biotin molecule (Fig. 1a). Biotin incorporation onto DL4-Fc was confirmed by Western Blot (Supplementary Fig. 1a) and its subsequent conjugation to streptavidin (SA)-coated polystyrene beads (SA-beads) was assayed by flow cytometry (Fig. 1b). Increasing amounts of biotinylated DL4-Fc were incubated with a constant number of SA-beads, and 0.5 µg of biotinylated DL4-Fc per $2 \times 10^5$ 25 µm beads appeared to saturate available binding sites.

To evaluate the influence of bead size on triggering Notch signaling, biotinylated DL4-Fc was conjugated to SA-beads of various sizes (6.5–100 µm). Functionalized beads were incubated with a reporter cell line, 3T3N1Cluc cells, in which Notch activation leads to luciferase expression. The design of the experiment ensured that the number of beads in each condition represented, in aggregate, the same total surface area and

therefore, the same number of DL4-Fc molecules presented to 3T3N1Cluc cells (Fig. 1c). Of note, the size of DL4-µbeads significantly affected their ability to activate Notch signaling, with 5 to 10-fold higher levels of luciferase reporter activity achieved with 25 µm DL4-µbeads, as compared to the other smaller and larger beads, or plate-bound DL4-Fc.

We next asked whether 25 µm DL4-µbeads, with the addition of IL-7, Flt3L, and SCF, could induce T cell development from mouse HSPCs. Lineage$^-$Sca1$^+$cKit$^+$ cells were isolated from E15 fetal livers and cultured with DL4-µbeads for 7 days. Imaging revealed a close interaction of HSPCs with DL4-µbeads (Supplementary Fig. 1b). To determine the optimal bead to cell ratio for the induction of T cell development, we tested a range from 1:1 to 27:1 (bead:cell) in a 7 day culture. Flow cytometric analysis showed the appearance of T-lineage CD90$^+$ and CD25$^+$ cells at all bead:cell ratios, with the 9:1 ratio yielding the highest frequency of T-lineage cells (Fig. 1d). As expected, unconjugated µbeads (UN) failed to support the differentiation of T cells and allowed for the appearance of CD11b$^+$ myeloid and CD19$^+$ B cells. In contrast, myeloid and B-lineage cell differentiation were inhibited in the presence of DL4-µbeads (Fig. 1d). Extending the culture period, with weekly bead and media changes, led to the appearance of CD4$^+$ CD8$^+$ double-positive (DP) cells, and αβTCR/CD3$^+$ CD4$^-$CD8$^+$ single-positive (SP8) cells by 28 (Fig. 1e), demonstrating the ability to generate T cells from mouse HSPCs in a serum- and stromal cell-free system.

**Human progenitor T cell development from CB and mPB.** We tested whether DL4-µbeads could induce T cell development from human CD34$^+$ HSPCs derived from umbilical cord blood (CB-CD34$^+$) and granulocyte-colony stimulating factor (G-CSF)-mobilized peripheral blood (mPB-CD34$^+$). Sorted CB-CD34$^+$ cells were cultured in serum-free media with DL4-µbeads (9 beads to 1 cell ratio) in the presence of cytokines for 7 days, which led to the emergence of T-lineage cells expressing CD34, CD7 and CD5 (Fig. 1f, g, Supplementary Fig. 2). We also evaluated the use of magnetic DL4-µbeads to induce T cell development from CB-CD34$^+$ cells while also facilitating bead removal by autoMACS separation (Supplementary Fig. 3). Extending DL4-µbead cultures for another 7 days resulted in a 400-fold total cell expansion and developmental progression to the CD34$^-$CD7$^+$ CD5$^+$ stage, indicating T-lineage commitment (Fig. 1f–h). Culturing mPB-CD34$^+$ cells with DL4-µbeads yielded a similar developmental progression, albeit with a lesser fold expansion after 14 days (Supplementary Fig. 4a–c). Both CB- and mPB-CD34$^+$ cells generated CD56$^+$ innate lymphoid cells (ILC) and CD33$^+$ myeloid cells early on (D7), but later on (D14), T-lineage cells made up the vast majority of the live hematopoietic cells (70% CD7$^+$ CD5$^+$) (Fig. 1g, Supplementary Fig. 4b).

Our findings show that DL4-µbeads can induce T cell development from both CB- and mPB-CD34$^+$ cells. We sought to determine whether the DL4-µbead system was able to fully capture the intrinsic T-lineage potential of these HSPCs. To address this, CB- and mPB-CD34$^+$ cells were placed in a limiting dilution assay (LDA) with DL4-µbeads for 14 days, and scored for the presence of CD45$^+$ CD7$^+$ CD56$^-$ cells. The results shown in Fig. 1i and Supplementary Table 1 demonstrate that CB-CD34$^+$ cells gave rise to T-lineage cells at a frequency of 1 in 15, while mPB-CD34$^+$ cells had a frequency of 1 in 172. Of note, the obtained T cell progenitor frequency of CB-CD34$^+$ cells cultured with DL4-µbeads is similar to that observed using the OP9-DL4 co-culture platform[22], validating the DL4-µbead system as an effective readout of T cell progenitor frequencies and underscoring the differences between CB- and mPB-CD34$^+$ cells to give rise to T cells.

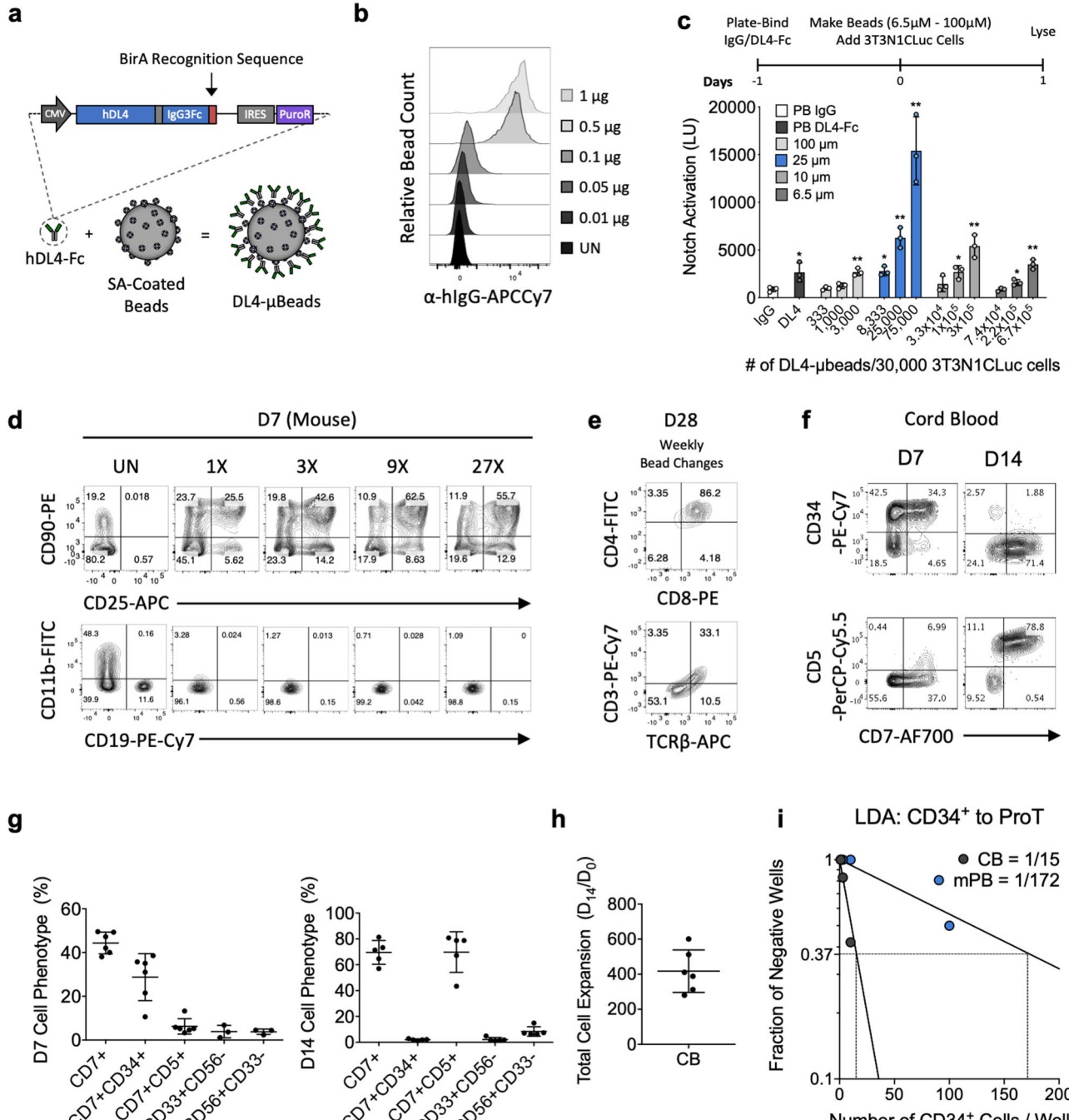

**Scalability and comparison of DL4-μbeads to plate-bound DL4-Fc cultures**. To extend the comparison of the DL4-μbead system to existing stromal cell-free cultures for the generation of T-lineage cells, we performed side-by-side comparisons using CB-CD34$^+$ cells cultured with DL4-μbeads along with plate-bound DL4-Fc (Fig. 2a). To directly compare the effectiveness of DL4-Fc when used in either platform, increasing amounts of DL4-Fc were adsorbed onto tissue culture-treated plates (1 μg/mL to 9 μg/mL), with 1 μg/mL being equivalent to amount bound onto the DL4-μbeads and 9 μg/mL being equivalent to what has been established in the literature. Cultures with DL4-μbeads gave rise to CD7$^+$ CD5$^+$ cells at a similar frequency and cell yield as that observed with DL4-Fc adsorbed at 3 and 9 μg/mL. Of note, cultures with DL4-μbeads significantly outperformed those with DL4-Fc adsorbed at 1 μg/mL. Additionally, an LDA analysis of T

cell progenitor frequency performed with DL4-μbeads or DL4-Fc adsorbed at 3 and 9 μg/mL showed a trend for a higher frequency outcome when assayed on DL4-μbeads (Fig. 2b and Supplementary Table 2). To establish the potential for scalability of the DL4-μbead platform, we took advantage of the Gas Permeable Rapid Expansion (G-Rex) system, a good manufacturing practice (GMP)-compliant bioreactor. $2.5 \times 10^4$ CB-CD34$^+$ cells were cultured with DL4-μbeads (9 beads to 1 cell ratio) for 7 days, followed by a full-media change and re-addition of fresh DL4-μbeads for another 7 days. After 14 days of culture, flow cytometric analysis and cell counts revealed a similar phenotype and fold expansion of cellularity, respectively, as to what was observed in tissue-culture treated 96-well plates (Fig. 2c), demonstrating the adaptability of the DL4-μbead platform to large-scale G-Rex systems.

**Fig. 1 DL4-µbeads activate Notch signaling and initiate the T cell program in HSPCs. a** Depiction of DL4-Fc fusion construct showing its components of the extracellular domain of human DL4, the Fc region of human IgG3 (IgG3Fc) and a BirA recognition sequence (AviTag) at its C-terminus. **b** Quantification of saturation of beads by DL4-Fc. DL4-µbeads were prepared by conjugating increasing amounts of DL4-Fc (0.01 µg to 1 µg) to $2 \times 10^5$ SA-coated µbeads and stained with anti-human IgG-APCCy7. **c** Evaluation of the effect of bead size on Notch activation. $3 \times 10^4$ 3T3-N1Cluc cells were incubated on plates pretreated with hIgG as negative control or with DL4-Fc at 10 µg/mL. The number of beads in each condition presented in aggregate, the same surface area and therefore, the same total number of DL4-Fc molecules. 24 h after plating, the cells were lysed and analyzed for luciferase activity ($n = 3$). **d**, Analysis of the ability of DL4-µbeads to support T cell development. Mouse fetal liver derived HSPCs were incubated for 7 days with unconjugated µbeads or DL4-µbeads in serum-free media containing SCF, IL-7 and Flt3-L. Co-cultures were harvested and analyzed for the presence of CD25, CD90, CD19 and CD11b using flow cytometry. **e** D28 cultures of DL4-µbeads and mouse LSKs were analyzed for the presence of late T cell markers (CD4, CD8, CD3 and TCRαβ). **f** Representative flow cytometry plots of human cord blood-derived CD34$^+$ cells cultured for 7 and 14 days with DL4-µbeads in serum-free media supplemented with SCF, IL-7 and Flt3-L. Cells were harvested and analyzed for the surface expression of CD34, CD5, and CD7 using flow cytometry. **g** Frequencies of the indicated proT-cell phenotypes after 7 days or 14 days, for the markers shown in (f) as well CD33 and CD56 (not shown) ($n = 6$). **h** Total cell expansion on D14, normalized to input day 0 CD34$^+$ CB-derived CD34$^+$ after culture with DL4-µbeads ($n = 6$). **i** MACS-enriched CB- and mPB-CD34$^+$ cells were placed in limited numbers in wells of a 96-well plate containing DL4-µbeads and cultured for 14 days before harvesting for flow cytometry. Individual wells were scored for the presence of T cell progenitors based on CD45$^+$ CD7$^+$ CD56$^-$. Statistical analysis was performed via the method of maximum likelihood applied to the Poisson model, and shown in Supplementary Table 1. Data represent means ± s.d. of $n$ independent experiments. SA = streptavidin. PB = plate-bound. UN = unconjugated. ($P > 0.05$); *$P < 0.05$; **$P < 0.01$ (two-tailed unpaired $t$-test).

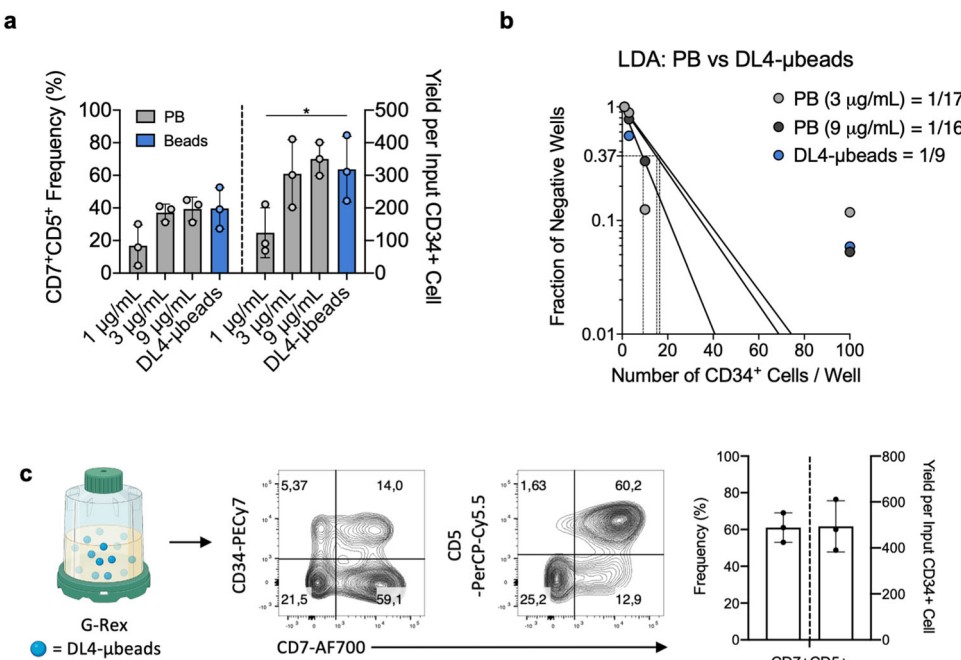

**Fig. 2 Scalability and comparison of DL4-µbeads to plate-bound DL4-Fc cultures. a** $1 \times 10^3$ CB-CD34$^+$ cells were cultured with DL4-µbeads or on plate-bound DL4-Fc, as indicated, for 14 days and analyzed by flow cytometry for the expression of CD5 and CD7. Cell counts were normalized to input day 0 CB-derived CD34$^+$ cells ($n = 3$). **b** CB-CD34$^+$ cells were placed in a limiting dilution assay for 14 days before scoring for the presence of CD45$^+$ CD7$^+$ CD56$^-$ cells. Statistical analysis was performed via the method of maximum likelihood applied to the Poisson model. **c** DL4-µbeads were assessed for adaptability to G-Rex system. $2.5 \times 10^4$ CB-CD34$^+$ cells were cultured with DL4-µbeads in a G-Rex 24-well plate format for 7 days, followed by a full-media change and re-addition of fresh DL4-µbeads for another 7 days. D14 cultures were harvested, counted and analyzed for the expression of CD34, CD7 and CD5, as indicated, from CD45$^+$-gated cells. Cell counts were normalized to input day 0 CB-derived CD34$^+$ cells ($n = 3$). PB = plate-bound. Data represent means ± s.d. of $n$ independent experiments. ($P > 0.05$); *$P < 0.05$ (one-way ANOVA followed by Fisher's Least Significant Difference). G-Rex image was created with BioRender.com (Toronto, ON).

**Differentiation of human PSCs to T-lineage cells by DL4-µbeads.** Despite advances in the ability to differentiate human PSCs into different hematopoietic lineages, the generation of T cells has been especially challenging, under serum- and stroma-free conditions. To address this, we induced the differentiation of human iPSCs as embryoid bodies (EBs) in chemically defined media with a stage-specific combination of morphogens and hematopoietic cytokines (Fig. 3a). After 8 days of differentiation, CD34$^+$ cells were harvested and stained for the presence of markers that identify arterial (CD184$^+$) and venous (CD73$^+$) endothelium (Fig. 3b). Cells lacking the expression of these two markers include a definitive hemogenic endothelium

population with T cell potential[23]. There is no apparent increase in cellularity from the iPSC to EB differentiation step, which given the percentage of CD34$^+$ cells obtained from EBs post-MACS enrichment, resulted in an approximate cell yield of 1 iPSC-derived CD34$^+$ cell per 10 iPSCs used to initiate the culture.

Sorted (D8) iPSC-derived CD34$^+$ cells (iPSC-CD34$^+$) were cultured with DL4-µbeads, and flow cytometric analysis of early time points, day 7 and 14, showed the appearance of CD34$^+$ CD7$^+$ cells, followed by CD34$^-$ CD7$^+$ CD5$^+$ cells (Fig. 3c,d). LDA analysis of iPSC-CD34$^+$ cells in culture with DL4-µbeads revealed a T cell progenitor frequency of 1 in $1.3 \times 10^3$ (Fig. 3e), which is

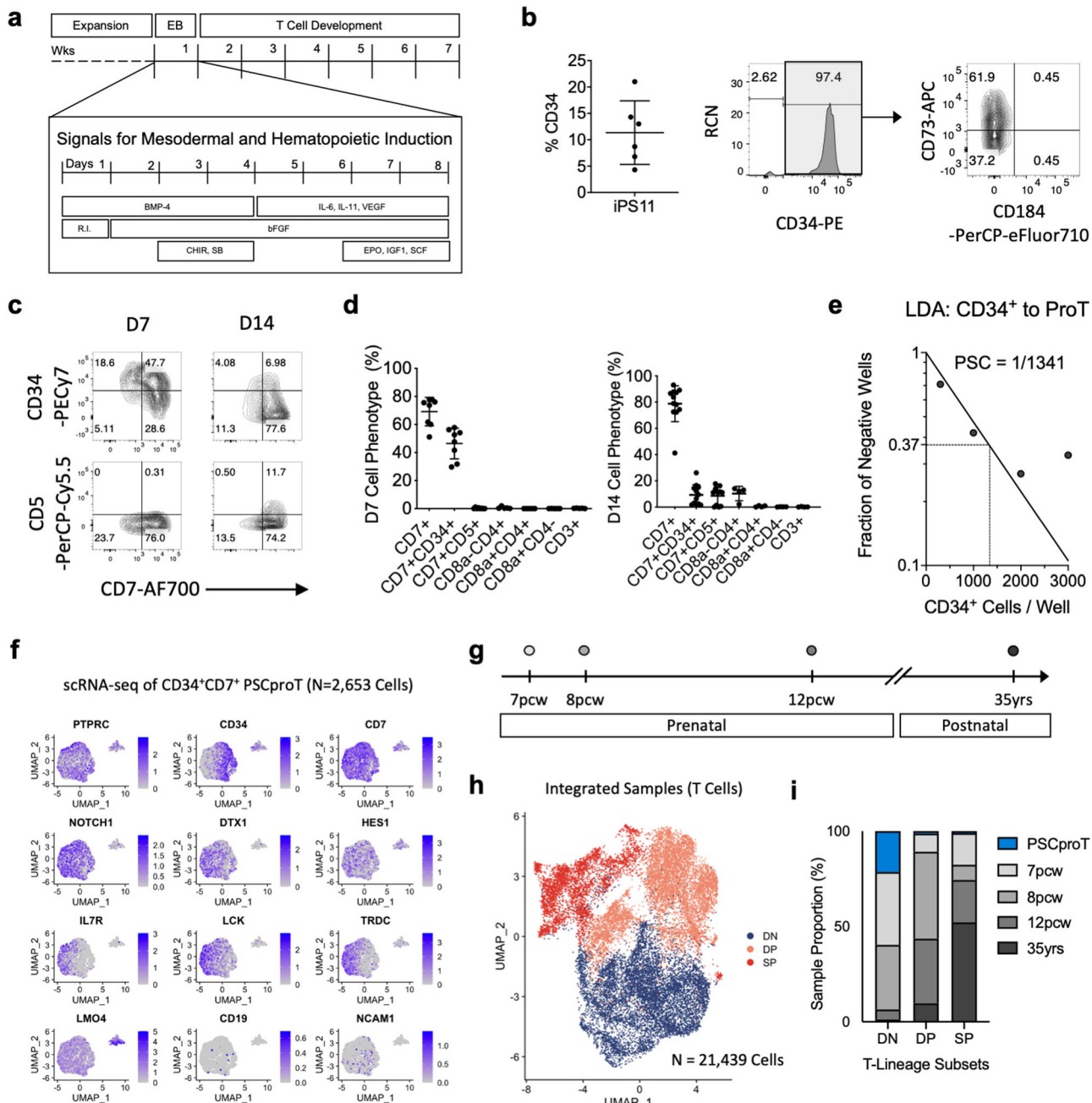

**Fig. 3 T cell development from human pluripotent stem cells using DL4-μbeads. a** Differentiation scheme used for hematopoietic induction and T cell development of human PSCs. EBs were generated from IPS11 cells during the first 24 h of culture and differentiated in chemically defined media with an optimized, stage-specific combination of BMP-4, bFGF, and VEGF together with hematopoietic cytokines as indicated. **b** Proportion of CD34+ cells at D8 harvest before CD34+ MACS-enrichment. Surface expression of CD184 and CD73 on CD34+ -gated cells post CD34+ MACS-enrichment (n = 6). **c** Representative flow cytometry plots of human iPSC-derived CD34+ cells cultured for 7 and 14 days with DL4-μbeads in serum-free media. Cells were harvested and analyzed for the surface expression of CD34, CD5, and CD7 using flow cytometry. **d** Frequencies of the indicated proT-cell phenotypes after 7 days or 14 days, for the markers shown in **c** as well CD3, CD4 and CD8 (not shown) (n = 6). **e** Limiting dilution assay of iPSC- CD34+ cells. MACS-enriched iPSC-CD34+ cells were placed in limited numbers in wells of a 96-well plate containing DL4-μbeads and cultured for 14 days before harvesting for flow cytometric analysis. Individual wells were scored for the presence of T-lineage cells based on CD45+ CD7+ CD56−. Statistical analysis was performed via the method of maximum likelihood applied to the Poisson model, and shown in Supplementary Table 1. **f** Single-cell RNA sequencing of 2,653 D7 iPSC-proT cells (GSE169279). UMAP visualization of the expression of genes from iPSC-proT sequencing data associated with early T cell development, Notch activation and alternative lineages. **g** Summary of the gestational age of human thymus single-cell RNA sequencing data retrieved from public repository (ArrayExpress: E-MTAB-8581). **h** UMAP visualization of integrated and batch-corrected sequencing data from PSC-proT and primary human thymocytes. Cell types are colored and annotated (DN = double-negative T cells; DP = double-positive T cells; SP = single-positive T cells). **i** Proportion of each sample within each T-lineage subset. Data represent means ± s.d. of n independent experiments.

significantly lower than what was observed for CB- and mPB-CD34[+] cells (Fig. 1i, Supplementary Table 1).

To dissect the cellular heterogeneity and transcriptional profile of iPSC-proT cells, we performed single-cell RNA sequencing on 2,653 CD45[+] CD34[+] CD7[+] iPSC-proT cells, sorted from DL4-µbead cultures (Fig. 3e, GSE169279). Uniform manifold approximation and projection algorithm was used to visualize the expression of T-lineage markers (*IL7R, LCK, TRDC*), Notch-target genes (*CD7, DTX1, HES1*) and alternative lineages (*LMO4, CD19, NCAM1*). Consistent with our flow cytometry results, there is a clear developmental progression of CD34[+] cells to more specified progenitors, which correspond with an increase in T-lineage transcription factors. A small cluster of less differentiated cells was present and marked by the expression of *LMO4*, likely representing group 2 innate lymphoid cells (ILC2)[24].

We sought to determine the transcriptional fidelity of iPSC-proT cells by comparing them to primary thymocytes derived from human prenatal (7pcw, 8pcw, 12pcw) and postnatal thymi (35 yrs)[25] (Fig. 3g, ArrayExpress: E-MTAB-8581). Sequencing data from a total of 38,619 cells were integrated and batch-corrected, and differential expression of genes between clusters were used to identify and annotate hematopoietic and thymic stromal cells (Supplementary Fig. 5). Using canonical T-lineage marker genes and recently defined developmental markers (*ST18* for DN, *AQP3* for DP and *TOX2* for DP-to-SP transition)[25], the T cell cluster was further divided into DN, DP and SP sub-clusters (Fig. 3h and Supplementary Fig. 6). The cellular composition based on the developmental stages showed that iPSC-proT cells were found to be present almost exclusively in the DN subset (Fig. 3i). Further analysis on genes that mark early T cells (*CD34, IGLL1*), migration (*CCR7*) and ontogeny (*LIN28B, HMGA2, IG2BP3*) suggest a developmental similarity between fetal and iPSC-T cell development (Supplementary Fig. 7).

**Generation of PSC-derived CD3[+]αβ T cells**. Analysis of iPSC-DL4-µbead cultures at later time points revealed the presence of CD7[+] CD5[+] T-lineage cells, and the emergence of CD3[−]CD4[+] immature single positive cells after 21 days (Fig. 4a and Supplementary Fig. 8). By day 35, DPs appeared, and they predominated by day 42 of culture. Total cellularity peaked at day 28 with a 750-fold expansion over input. Subsequently, there was a decrease to 100-fold by D42, consistent with the cell death expected to occur with thymocyte selection events (Supplementary Fig. 8). Differentiation of iPSC-CD34[+] cells for 42 days showed consistent progression to CD3/αβTCR-expressing DP cells with a ~150 fold expansion over input iPSC-CD34[+] cells (Fig. 4b). We also examined T cell development from STIPS.A3, an iPSC cell line reprogrammed from T cells (Fig. 4a). STIPS.A3 displayed rapid and orderly T cell development, with an accelerated CD3/αβTCR expression by D21. These findings, together with our results from mouse and human HSPCs, demonstrate that DL4-µbead cultures have a robust capacity for generating T-lineage cells from multiple sources of stem cells.

**DL4-µbead-derived human proT cells show thymus-reconstituting ability in vivo**. We sought to assess the ability of CD34[+] CD7[+] proT cells derived from HSPC/DL4-µbead cultures to engraft the thymus of NSG mice. The production of proT cells was scaled up, switching from 96-well plates to T75 flasks, again demonstrating the scalability of the DL4-µbead system. CD34[+] CD7[+] proT cells were sorted from day 7 cultures and injected intrahepatically into neonatal NSG mice (Fig. 5a). After 4 weeks, flow cytometric analyses of recipient mice thymuses revealed high levels of engraftment (~80%) by human CD45[+] cells, which included DP and SP thymocytes, and lacked

myeloid and B-lineage cells (Fig. 5b). After 12 weeks post-engraftment, we detected mature αβTCR/CD3-bearing CD8[+] and CD4[+] T cells in the thymus and spleen, indicating that proT cells derived from DL4-µbeads were capable of reconstituting the periphery of immunodeficient NSG mice (Fig. 5c). To confirm whether the T cells present in the spleen were functionally mature, CD3[+] T cells were harvested after 12 weeks and stimulated with anti-CD3/CD28 in vitro (Fig. 5d). As anticipated, both IFN-γ and TNF-α were readily detected in stimulated T cells, indicating that human CD34[+] CD7[+] proT-cells produced by DL4-µbeads are capable of homing to and engrafting thymi in vivo and subsequently giving rise to mature cytokine-secreting CD3[+] T-cells in the periphery.

## Discussion

In this study, we present a bead-based platform for generating T-lineage cells from multiple sources of stem cells. Our design relies on the immobilization of DL4-Fc to µbeads of a determined size, obviating the need for stromal elements and providing a scalable cell suspension culture alternative to existing plate-bound approaches. Stromal cell-free strategies have until now relied on the immobilization of Notch ligands to the surface of tissue-culture plates[14–17,19], since DL4-Fc in soluble form fails to activate Notch signaling[26]. This approach has a practical limitation in terms of space required to generate clinically relevant numbers of T-lineage cells using plates[20,21]. In addition, plate-bound systems are not conducive to the dynamic nutrient mixing required to scale up production. Here, we took advantage of the linearly scalable G-Rex system[27], demonstrating the adaptability of the DL4-µbead platform. Moreover, a direct comparison of DL4-µbeads to plate-bound conditions showed comparable effectiveness in supporting T-lineage differentiation from HSPCs. However, DL4-µbeads appeared to be more efficient in inducing Notch-dependent signaling when equal amounts of DL4-Fc were used. Therefore, technologies that are amenable to suspension cultures and seamlessly incorporated into automated closed bioreactor systems are warranted. Our results indicate that the DL4-µbead system meets these requirements.

The presentation of Notch ligands on beads has been previously investigated[28]. However, their capacity to support the generation of T cells was not addressed. Our studies have identified several key features for optimal induction of Notch signaling using a bead-based system. First, we identified bead size as a key parameter, with a 25 µm diameter providing the highest levels of Notch activation across evaluated sizes. This explains previous studies where Notch-ligand functionalized beads of less than 10 µM were unable to induce the differentiation of HSPCs into T cells, but rather supported the differentiation of HSPCs into alternative non-T cell lineages[28]. Secondly, the bead to cell ratio was an important factor in optimizing Notch signaling. As DL4-µbeads lack the capacity for ligand endocytosis, it remains unclear how the necessary force of 4-12 pN that is generated to expose proteolytic cleavage sites in the Notch receptor[29–31]. One possibility points to the force created by the product of the bead's mass and acceleration, which is reached by a 25 µm bead and enables the activation of the Notch receptor. Further investigation could reveal the physical characteristics that enable this particular bead size and ratio to effectively turn on Notch signaling. Uncovering these requirements will allow for additional design parameters to further refine the technology. As T cell differentiation cultures in this study were static, it is possible that bead size will need to be re-evaluated in dynamic cultures as the biophysical properties change.

Human proT cells derived from HSPCs cultured with DL4-µbeads are capable of thymic engraftment, and therefore will

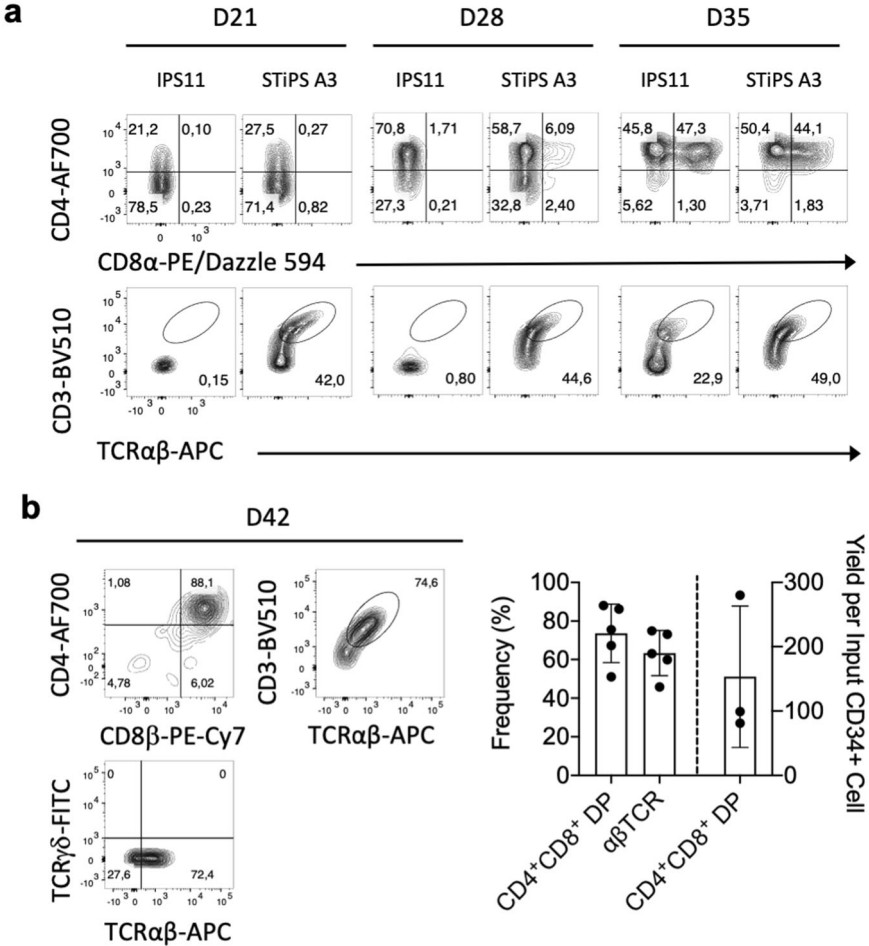

**Fig. 4 Generation of PSC-derived CD3$^+$αβ T cells. a** Representative flow cytometry plots of human IPS11- and STIPS.A3-derived CD34$^+$ cells cultured for up to 35 days with DL4-μbeads. Cells were harvested, counted and analyzed by flow cytometry for the surface expression of CD4, CD8, TCRαβ and CD3, as indicated. **b** D42 IPS11-derived T-lineage cells were harvested and stained for the expression of CD4, CD8β, TCRαβ, TCRγδ and CD3, as indicated. Frequency of CD4$^+$ CD8$^+$ DP cells and αβTCR-expressing cells are plotted, and total cell expansion was normalized to input day 0 PSC-derived CD34$^+$ cells after culture with DL4-μbeads for 42 days ($n = 3$). Data represent means ± s.d. of $n$ independent experiments. All data gated on DAPI$^-$CD45$^+$ cells. IPS11 = iPSC line derived from fibroblasts; STiPS A3 = iPSC line derived from T cells.

enable the development of cell-based therapies for immune regeneration. The development of cellular therapies relying on functionally validated, banked, broadly histocompatible cell types would have a major impact on the applicability and cost of adoptive T cell therapies[4,5]. This prospect raises the challenge of differentiating PSCs to T-lineage cells under serum- and stroma-free conditions. Here, we benchmarked iPSC-proT cells to publicly available human thymus scRNA-seq data[25], which equated these cells to early fetal thymocyte progenitors, rather than the progenitor counterparts present in the adult thymus.

The limited progression to mature SP stages is not surprising as the system lacks stromal-elements that would provide positively-selecting ligands. This could be overcome through stimulation of purified iPSC-derived CD4$^+$ CD8$^+$ DP cells with anti-TCR/CD3 antibodies[32–34]. Nonetheless, the differentiation of TCR-transduced iPSCs or re-differentiation of iPSCs from T cells enhances the efficiency of generating αβTCR-expressing T cells[32–34], which was also the case in our iPSC-DL4μbeads cultures.

The standardization of the DL4-μbead platform coupled with LDAs revealed intrinsic differences in T cell differentiation from different stem cell sources. We found a clear hierarchy from CD34$^+$ cells, with CB > mPB > iPSCs in their T cell progenitor frequency when cultured with DL4-μbeads. These findings are consistent with the observation that CB-CD34$^+$ cells are able to effectively give rise to T cells in vitro and in vivo[35,36]. Reports comparing mobilization regiments revealed a direct effect of G-CSF on mPB-CD34$^+$ cells, with the proportion of HSCs capable of reconstituting NSG mice being decreased[37]. The diminished capacity of iPSC-CD34$^+$ cells to support the differentiation of T-lineage cells could be attributed to the inability of existing protocols to fully recapitulate the spatial and temporal events occurring during embryonic hematopoiesis[38–41]. Strategies to bias lineage choice or generate true HSCs would increase the efficiency of T cell generation[42–44], adapting these approaches to the DL4-μbead platform is of interest and could increase the efficiency of T-lineage generation from PSC.

The DL4-μbead approach offers a simple, powerful, reproducible and potentially scalable platform for T cell generation. The modularity of the bead platform could allow for the attachment of other Notch-ligands or ligands that interact with mechanoreceptors[45]. This flexibility will facilitate the investigation of Notch signaling in other lineages and permit adoption of DL4-μbeads to multiple types of culture systems[46–48]. In summary, our platform has many experimental and clinical applications, which when combined with emerging genetic engineering strategies should facilitate the development of new cell-based immunotherapeutic approaches.

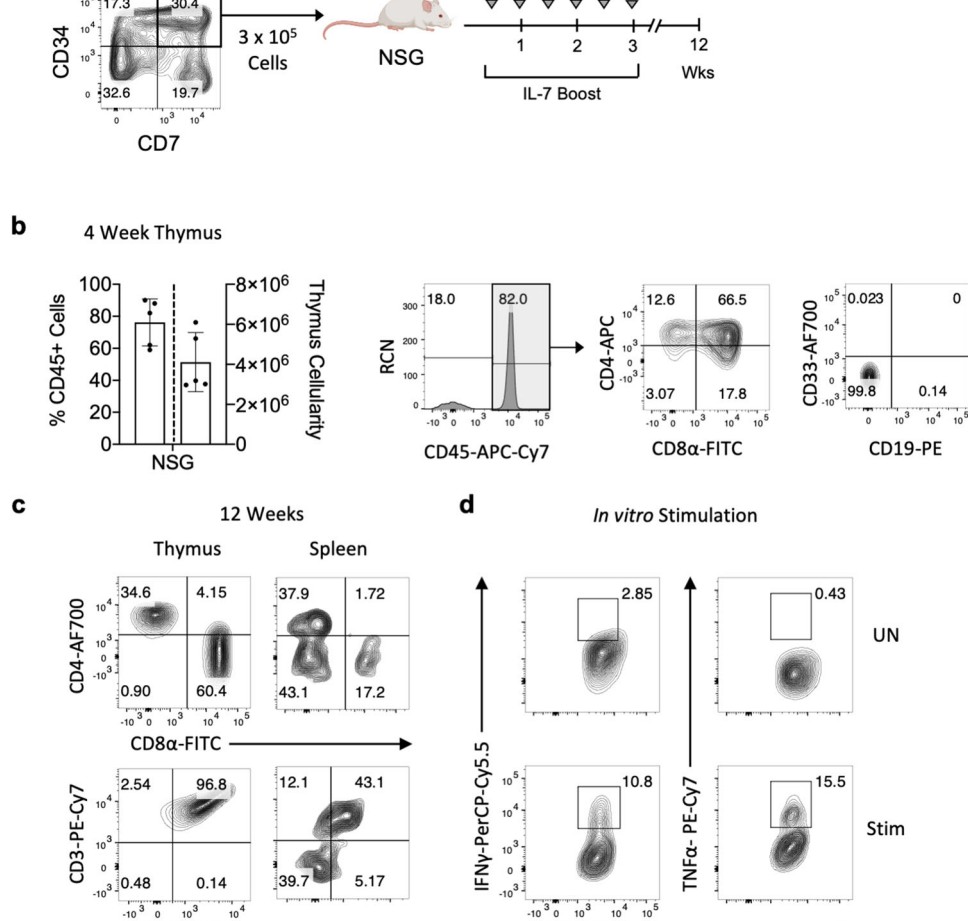

**Fig. 5 Engraftment and differentiation of human proT cells in NSG mice. a** Schematic of CB-derived proT into neonatal NSG mice. CD34[+] cells derived from human CB were incubated with DL4-µbeads for 7 days. CD34[+] CD7[+] progenitor T (proT) cells were sorted using flow cytometry and $3 \times 10^5$ were injected intrahepatically into neonatal NSG mice. Human IL-7/M25 injection boosts were given at 3-4 day intervals. **b** Percentage of live human CD45[+] cells in mouse thymi 4 weeks post-injection and quantification of total thymic cellularity ($n = 5$ mice). Representative flow cytometry plots of 4-week engrafted thymuses. Lineage analysis is on gated CD45[+] -cells using the cell surface markers CD4, CD8, CD33 (myeloid) and CD19 (B cell). **c** Representative flow plots of CD4, CD8, CD3 and TCRαβ expression on human CD45[+] -gated cells harvested from thymus and spleen 12 weeks post-injection ($n = 3$). **d** Representative flow plots for intracellular IFN-γ, and TNF-α upon in vitro stimulation of human CD45[+] CD3[+] cells. Cells were harvested from the spleen after 12 weeks, and activated with Immunocult Human CD3/CD28 T Cell Activator for 3 days, prior to staining for intracellular cytokines. Data represent means ± s.d. of $n$ independent experiments. Mouse image is from BioRender.com (Toronto, ON).

## Methods

**Mice**. NOD.cg-*PrkdcscidIL2rg*tm/Wjl/Sz (NOD/SCID/γc[null] [NSG]) and C57BL/6 mice were purchased from Jackson Laboratory (Bar Harbor, ME), housed, bred and maintained in the Comparative Research Facility of the Sunnybrook Research Institute, under specific pathogen-free conditions. All animal procedures were approved by the Sunnybrook Research Institute Animal Care Committee and performed in accordance with the committee's ethical standards.

**Human CD34[+]HSPC samples**. Human UCB samples were obtained by syringe extraction and collected in a blood-pack unit containing citrate phosphate dextrose anti-coagulant (Baxter Healthcare, Deerfield, Illinois) from consenting mothers following delivery in accordance to approved guidelines established by the Research Ethics Board of Sunnybrook Health Sciences Centre. Within 24 h of collection, cord blood mononuclear cells were isolated using Ficoll density centrifugation, and pre-enriched for CD34[+] cells using magnetic-activated cell sorting (MACS), as previously described[22]. Sorted human HSCs (CD34[+]) were greater than 90% pure as determined by post-sort flow cytometry analysis. MACS-purified CD34[+] human GCSF-mPB samples were obtained from StemCell Technologies (Vancouver, BC Canada).

**Cell lines**. HEK-293T and NIH3T3 cells were purchased from American Type Culture Collection (ATCC, Manassas, VA) and were maintained in DMEM [supplemented with 10%(v/v) FBS, 2 mM Glutamax, Penicillin (100 U/ml)/

Streptomycin (100 mg/ml) (Thermo Fisher Scientific), 2 mM 2-mercaptoethanol (Sigma-Aldrich)].

**DL4-Fc protein design and purification**. The coding sequence of the extracellular domain of human DLL4 was cloned upstream of a Histidine (His) tag followed by the Fc portion of human IgG3 (including the hinge region) along with a BirA recognition sequence (Avitag) at the C-terminus interspaced by Gly-Ser spacers (DL4-Fc, Fig. 1a). This construct was inserted into a pIRESpuro2 mammalian expression plasmid (Clontech, CA). The resulting plasmid was transfected into HEK-293T cells using a standard CaPO4 transfection method and cells with a stably integrated plasmid were selected based on puromycin resistance (2 mg/mL). Cells were expanded in Freestyle 293 expression media (Thermo Fisher Scientific). Supernatant containing the HEK-293T cell-secreted DL4-Fc fusion protein was subjected to fraction purification using HiTrap Protein G affinity columns (GE Healthcare) attached to the ÄKTAprime plus (GE Healthcare) automated chromatography system. Eluted fractions that contained DL4-Fc protein, as determined by Western blot analysis, were pooled and protein concentration was determined by spectrophotometry.

**Biotinylation of DL4-Fc**. A single biotin molecule was enzymatically conjugated to the AviTag sequence at the C-terminus of DL4-Fc using the BirA-500 Kit (Avidity Biosciences, La Jolla, CA) according to the manufacturer's instructions. Briefly, 2.5 µg of BirA was added to 500 µg of DL4-Fc in a reaction volume of 500 µL PBS, containing free biotin, and incubated for 1 h at RT. To remove any remaining free

biotin, the reaction material was desalted using a 40 K MWCO Zeba™ Spin Desalting Column (Thermo-Fisher) according to the manufacturer's instructions. Biotinylated DL4-Fc was subsequently stored at 4 °C.

**DL4-Fc plate-bound preparation**. DL4-Fc was diluted in chilled phosphate-buffered saline (PBS) to the indicated concentration and 50 µL/well was used to coat flat-bottom standard tissue-culture 96-well plates overnight at 4 °C. Wells were washed once with PBS before seeding cells to remove any unbound ligand from the wells.

**Generation of DL4-µbeads**. 1 µg of biotinylated DL4-Fc was incubated with streptavidin (SA)-coated polystyrene or iron-oxide-coated µbeads (Spherotech, Lake Forest, IL), ranging from 1 to 100 µm in diameter, for 30 min at RT. DL4-µbeads were washed with 4 mL PBS and spun down at $3000 \times g$ for 10 min. After a second wash, DL4-µbeads were re-suspended in various volumes of PBS to achieve the indicated concentrations. Unconjugated µbeads were prepared in parallel, to serve as a negative control.

**Notch reporter assay**. RBPJ consensus binding sites (6X) were inserted into the promoterless pGL4.17[luc2/Neo] plasmid (Promega, Madison, WI), yielding a Notch activation luciferase reporter plasmid, pGL4.17-N1Rep, which also confers resistance to Neomycin. NIH3T3 cells were co-transfected with pMIGR-NOTCH1 (gift from Warren Pear, University of Pennsylvania, PA) and pGL4.17N1Rep plasmids. Cells that were resistant to Neomycin treatment (1 µg/mL) were selected and sorted for GFP expression. NIH3T3 cells were single-cell sorted into 96 well plates. The resulting clone, named 3T3N1CLuc, was selected based on high luciferase activity following Notch activation and low background luciferase activity in the absence of Notch signaling.

DL4-µbeads were added to $3 \times 10^4$ 3T3N1CLuc cells in a standard tissue culture-treated flat-bottom 96-well plate and incubated overnight in αMEM supplemented with 5% FBS. Cells were lysed and assayed for luciferase activity using the Firefly Luciferase Assay Kit 2.0 (Biotium, Fremont, CA) according to the manufacturer's instructions. Briefly, growth medium was removed and cells were washed with PBS prior to adding lysis buffer. Cells were lysed by freezing at −80 °C for 10 min followed by thawing before transferring lysates to a 96-well flat-bottom opaque polystyrene plate (Corning, Corning, NY). D-luciferin was prepared and added to each well by automatic dispenser and analyzed using the Synergy H1 plate reader (BioTek Instruments Inc., Winooski, VT). Human IgG and DL4-Fc plate-bound controls were prepared the previous day by adsorbing 50 µL/well of protein (10 µg/mL) to flat-bottom 96-well plates overnight at 4 °C prior to washing and seeding of 3T3N1CLuc cells the following day.

**T Cell Differentiation with DL4-µbeads**. Human HSPCs (CB CD34+, mPB CD34+ and iPSC CD34+) were cultured with DL4-µbeads at a starting ratio of 1 cell to 9 beads (1:9) in StemSpan SFEM II supplemented with StemSpan T Cell Progenitor Expansion Supplement (StemCell Technologies) at 37 °C in 5% CO2 incubator. Mouse FL LSKs were cultured with DL4-µbeads at the indicated ratios in IMDM + BSA, insulin, transferrin (BIT) (StemCell Technologies) supplemented with 100 ng/mL each of FMS-like tyrosine kinase 3 ligand (Flt3L), and stem cell factor (SCF) and interleukin (IL)-7. After 7-day culture intervals, cells were counted and subjected to a full media change, including re-addition of DL4-µbeads at a 1:9 ratio.

**Progenitor T cell differentiation in G-Rex system**. CB-CD34+ cells were cultured with DL4-µbeads in the G-Rex 24 Multi-Well Cell Culture Pate (Wilson Wolf). $2.5 \times 10^4$ CB-CD34+ cells were cultured with DL4-µbeads at a starting ratio of 1 cell to 9 beads in StemSpan SFEM II supplemented with StemSpan T Cell Progenitor Expansion Supplement (StemCell Technologies) at 37 °C in 5% CO2 incubator. After 7 days, a full-media change was performed followed by re-addition of fresh DL4-µbeads for another 7 days.

**Limiting dilution assay**. Limiting dilution assay (LDA) was performed by serial dilutions of CD34+ cells sourced from CB, mPB and iPSC and cultured with DL4-µbeads. CD34+ cells were sorted using the FACSAria cell sorter, and 1 ($n = 24$), 3 ($n = 12$), 10 ($n = 12$), 100 ($n = 12$), 1000 ($n = 6$) or 3000 ($n = 6$) were directly deposited into individual wells of a 96 well/plate containing DL4-µbeads. Cells were cultured for 7 d, and individual wells were harvested and analyzed by flow cytometry for the presence of CD45+ CD7+ CD56− cells. Progenitor frequencies were determined by the method of maximum likelihood applied to the Poisson model[49].

**Adoptive transfer of ProT cells into immunodeficient mice**. $2 \times 10^5$ CD34+ HSPCs were incubated with $1.8 \times 10^6$ DL4-µbeads in T75 flasks (Thermo Scientific) for 7 d, as indicated above. CD34+ CD7+ proT cells were sorted by flow cytometry and $3 \times 10^5$ cells, prepared with a complex of rhIL-7 (0.5 µg/mouse) and anti-IL-7 monoclonal antibody (mAb), clone M25, (2.5 µg/mouse) in a total volume of 30 µl in PBS, were injected intrahepatically into day 3–5 NSG neonates. Mice were boosted with IL-7/M25 complex every 3–4 days for 3 weeks. Thymus, spleen and bone marrow were harvested at 4 and 12 weeks after intrahepatic transplant. Single cell suspensions were counted, stained and analyzed by flow cytometry.

**Stimulation of T cells**. Splenocytes were harvested from NSG mice 12 weeks after intrahepatic injection of DL4-µbead- derived proT cells. CD3+ cells were sorted by flow cytometry and activated with Immunocult Human CD3/CD28 T Cell Activator in ImmunoCult-XF T Cell Expansion Medium (StemCell Technologies) for 3 days, following manufacturer's protocol. Stimulated cells were washed post-stimulation and stained for intracellular cytokines.

**Maintenance and differentiation of iPSCs**. Human iPS11 (Alstem Cell Advancements, Richmond, CA) and STiPS A3 (HSCI iPS Cell Core Facility, Boston, MA) were cultured and maintained in mTeSR plus complete medium (StemCell Technologies) on Matrigel (BD Biosciences). IPS11 cell line was used to generate all PSC-related data unless indicated otherwise. To generate self-aggregated EBs, iPSCs were gently dissociated with ReLeSR and small clusters were formed using a cell scraper. The cell aggregates were transferred to Ultra-Low Attachment 6-well plates (Corning) in StemPro34 (Invitrogen) media containing BMP-4 and ROCK inhibitor (R&D systems). The EB media was changed after 24 h to StemPro34 in the presence of BMP-4 and bFGF for the next 18 h, and then in the presence of BMP-4, bFGF, SB-431542 and CHIR-99021 for the following 48 hr (days 2–4). At day 4, EBs had the appearance resembling crumpled balls of paper. Media was replaced to StemPro34 containing VEGF, IL-6, IL-11, and bFGF (all cytokines and growth factors from R&D Systems). At day 6, current media was topped off with StemPro34 containing VEGF, IL-6, IL-11, IGF-1, SCF, EPO and bFGF. Cultures were maintained in a hypoxic environment at 5% CO2/5% O2/90% N2 for 8 days. On day 8, EBs were dissociated, and the harvested cells were MACS-enriched for CD34+ cells and incubated with DL4-µbeads plus StemSpan SFEM II (Stemcell Technologies) supplemented with StemSpan T Cell Progenitor Expansion Supplement (Stemcell Technologies) as described above, in order to generate T cells.

**Single-cell RNA sequencing data analysis**. iPSC CD34+ cells were cultured with DL4-µbeads for 7 days, at which time CD45+ CD34+ CD7+ cells were sorted for single cell RNA sequencing using 10x Genomics. A total of 2653 were sequenced and the resulting data were aligned and quantified using the CellRanger 3.1 pipeline (3' v3 chemistry) using the GRCh38 human reference genome. Sequencing data (GSE169279) were analyzed using R software version 4.0.0 with the package Seurat, version 3.1.

Sequencing data from PSC-proT cells were integrated with human thymic datasets downloaded from a public repository. Three prenatal and one postnatal datasets were downloaded from ArrayExpress (E-MTAB-8581) as FASTQC files: 7pcw- CD40, 8pcw-CD41, 12pcw-F45 and 35yrs-A43, and processed using the same pipeline.

A total of five datasets were merged, integrated and batch-corrected using Harmony. Each dataset was normalized and scaled separately using the SCTransform wrapper in Seurat and then merged. To reduce the impacts of cell cycle on cell clustering and dimensionality reduction, a custom set of cell cycle genes were regressed before further analysis. Cell subsetting based on gene expression was done to identify and analyze T-lineage clusters and differential gene expression between samples.

**Statistical analysis**. The data and error bars are presented as means ± s.d. To determine statistical significance between two independent groups, a two-tailed unpaired $t$ test was performed using GraphPad Prism 8. A one-way analysis of variance (ANOVA) was performed when comparing more than two groups. The ANOVA analysis was followed by Fisher's Least Significant Difference test for pre-planned comparisons. Statistical significance was determined as follows: not significant ($P > 0.05$); *$P < 0.05$; **$P < 0.01$.

**Reporting summary**. Further information on research design is available in the Nature Research Reporting Summary linked to this article.

## Data availability

RNA sequence data generated in this study have been deposited to GEO under the accession code GSE169279. Human thymic datasets were downloaded from a public repository: ArrayExpress (E-MTAB-8581) as FASTQC files. Source data is provided with this paper. Source data are provided with this paper.

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

## Acknowledgements

The authors thank G. Awong for expertize in cell sorting, C.R. Lee for expert technical support with animal experiments, and E. Herer and R. Kung from the Women and Babies Program at Sunnybrook Health Sciences Centre (Toronto, ON, Canada) for their ongoing support by providing umbilical cord blood samples. This work was supported by grants from the Canadian Institutes of Health Research (CIHR FDN-154332), the Canadian Cancer Society Research Institute, Ontario Institute for Regenerative Medicine, National Institutes of Health (1R01HL147584-01A1), Medicine by Design: a Canada First Research Excellence Fund Program at the University of Toronto, Stem Cell Network, and the Krembil Foundation. J.C.Z.-P. was supported by a Canada Research Chair in Developmental Immunology.

## Author contributions

A.C.T.G. designed and performed the experiments, analyzed all data, and wrote the manuscript; E.C.M. and P.M.B. maintained and differentiated PSC cultures; J.D.S.C, D.L., and J.H. performed some in vitro experiments; M.M. provided critical experimental advice and edited the manuscript; S.T. and M.K.A. provided experimental advice and edited the manuscript; and, J.C.Z.-P. designed experiments, analyzed the data, and wrote the manuscript.

## Competing interests

The authors have submitted a patent describing the method of generating cells of the T-lineage using DL4-μbeads. AC.T.G., M.M., and J.C.Z.-P. are co-founders and shareholders of Notch Therapeutics. The other authors declare no competing interests.
