## [Peer Review File · Nature Communications]

REVIEWER COMMENTS

Reviewer #1 (Remarks to the Author):

Many steps of T cell development can be recapitulated in vitro using stromal cells expressing Notch ligands, or by immobilizing Notch ligands on tissue culture plates. Trotman-Grant et al. refine an approach to generate mature T cells at large scale without supporting cells. Such a method will allow scalable production of T cells in a non-xenogenic stromal cell-free system that is compatible with clinical applications.

The authors use a bead-based method in which DL4-Fc is immobilized onto surface of microbeads that can activate Notch and supports differentiation of mouse and human hematopoietic stem and progenitor cells. Data shown in the paper suggest that the method of attaching DL4 to microbeads supports differentiation of hematopoietic progenitors to proT and mature T cells. In my opinion, the improvements are not clearly established, and new scientific insights are not provided in the present manuscript. It is likely that the new method is indeed scalable, and will allow some clinical applications. However, such applications are not yet demonstrated, and the work appears preliminary.

1. Cell-free systems to generate T cells have been described (PMID 29208547). It was earlier shown that using plate bound DLL4 Fc-fusion protein, adult HSPCs can efficiently generate in vitro cultured T-cell precursors that give rise to functional T-cell compartment in vivo. The authors did not compare their method with existing methods, and I am uncertain whether the bead-based method is more efficient than the plate bound method, although the beads are of course scalable. Data in Fig 1h showed bead-based method gives 400-fold expansion on d14 as compared to day0, but comparison to plate-bound methods is not performed.

2. The authors do not provide significant scientific insights. It is possible that investigation of why the highest activation of Notch signaling is seen with 25micron size bead and not smaller or larger than 25micron size may provide such insights, as the authors discuss, but these observations are at present not understood.

Minor points:

1. Supplementary Fig 2 is not cited in the text.
2. In line 157, authors likely mean Fig2f instead of Fig2e.
3. In Fig 2f, sc-RNA seq data appears to identify a small cluster of cells that expresses LMO4 and also expresses PTPRC and HES1. But the authors state in text line 162-163 that this small cluster is not Notch-responsive and characterized by lack of PTPRC. The authors should explain the discrepancy between text and data.

Reviewer #2 (Remarks to the Author):

The manuscript describes the development of a stroma-free system for inducing T lineage development from mouse and human hematopoietic stem and progenitor cells (HSPC) and human induced pluripotent stem cells (iPSC). The investigators have replaced the stroma component used in other systems by using microbeads that present DLL4 to the hematopoietic cells. They also use a commercially available serum-free medium that has been developed for T cell differentiation. The main message of the paper is that avoidance of the stroma cells will allow more simple clinical translation for adoptive cellular therapy. The data shows convincingly that the bead method induces T cell lineage commitment to the ProT stage but positive selection and maturation in vitro does not occur. The lack of data on FACS analysis, cell yield and function of iPSC derived cells limits interpretation of the technology.

Major revisions:

1. The following statement in the abstract is misleading :“ Here, we report a stromal cell-free, microbead-based approach that supports the efficient in vitro development of both human

progenitor T (proT) cells and mature T cells from CD34+ cells sourced from cord blood, GCSF-mobilized peripheral blood, and pluripotent stem cells (PSCs).” No evidence is given that mature T cells (SP4 or SP8) are generated from these populations in vitro. Mature SP cells are only shown in mouse thymus and spleen after in vivo transfer into NSG mice of human proT cells generated with beads from CB.

2. The statement in the abstract “DL4- μ beads, along with lymphopoietic cytokines, induced an ordered sequence of differentiation from CD34+ cells to CD34+CD7+CD5+ proT cells to CD3+ $\alpha\beta$ T cells should also be modified.

3. T cell differentiation is shown mostly as single flow cytometry profiles with insufficient information on how the profiles have been generated. The cell populations shown in the summary graphs do not add up to 100% of the cells in culture-what are the other phenotypes? In all FACS derived data please show how the profiles shown have been gated.

4. In vivo data is shown only as representative flow cytometry plots-summarized quantitative data should be given to understand reproducibility.

5. As the paper emphasizes the potential advantage of the bead system for scale-up capabilities, more detailed information on cell yield (total and each subpopulation) should be provided to get a sense of how far the process needs to be scaled to produce a therapeutic cell dose. Interesting information on the frequency of cells with CD7+ potential in different sources has been given, and fold increase in bulk cultures is shown, but it is unclear how many cells were used to initiate the cultures and how many were recovered i.e. how close are these cultures to the cell dose required for a patient? In the case of iPSC how many cells were produced at each step of the procedure?

e.g. iPSC to EBs to CD34+ to CD7+ and DP. Can the iPSC derived cells be expanded after culture? 6. Because the focus is on translation, it would be useful to know how this method compares to at least one of the existing stromal-based methods in terms of yield and differentiation, and also show some control data on cultures without beads.

7. Functional data in the paper is limited to cells recovered from the spleen of NSG mice after infusion of proT cells from CB cultures. It is important to know how the iPSC generated cells behave functionally. How do the cells compare to those from thymus and/or peripheral blood?

Minor revisions:

1. Authors show importance of bead size for notch signaling using the notch reporter cell line 3T3N1C. Are there any data on effect of bead size on T cell differentiation or yield from iPSC and/or HSPC?

2. Engraftment of NSG thymus by CB derived proT cells is shown as a frequency of human cells. As NSG mice contain little if any murine hematopoietic cells frequency of human cells will be high. How many human cells are recovered from each thymus?

3. LDA is given for “T cell Progenitor” output based on a definition of CD45+CD56-CD7+: does this define a progenitor or rather all T lineage cells?

4. Legends for LDAs (Fig 1i and Fig 2e) mention p values but the figures do not seem to show *

5. Please specify what iPSC line is being used in each figure -e.g. Supplem Fig 7 legend

6. Supplemental figures 5, 6 legends should explain the cell source for the UMAP data

7. Legend of Fig 3d should state method of stimulation.

Reviewer #3 (Remarks to the Author):

T cell based immuno-therapies rely largely on the expansion of autologous T cells. In vitro de novo generation of T cells from human progenitor cells for clinical applications is currently not established in part due to difficulties of producing large T cell numbers using non-xenogenic stroma cell free systems that would be compatible with clinical applications.

In this manuscript the authors report the development of a stromal-cell free, DL4 coated microbead-based system that allows in vitro development of both murine and human progenitor T cells and mature T cells.

The authors first optimized bead size for the DL4 bearing μ beads as well as bead to cell ratios for their stromal cell-free T cell differentiation system.

They used this system and tested different sources of human hematopoietic progenitor cells for their ability to efficiently differentiate into T cells, these included CD34+ cells derived from human

cord blood cells, GCSF-mobilized peripheral blood and human iPSCs. The frequency with which these cells differentiated into T cell progenitors differed between the cell source with following hierarchy: CB > mPB > iPSCs . Single cell RNA sequencing was used to characterize the T cell specific gene expression pattern and adoptive transfer of CD34+CD7+ pro T cells into NSG mice was performed to demonstrate that the in vitro generated T cell progenitors were able to efficiently engraft the thymus and to functionally mature into peripheral T cells.

This is a very interesting study and to a large extend very well performed. It has the potential to be further developed so that it is suitable for publication.

Major comments:

1. The optimization and identification of the optimal bead size and bead to cell ratio is very nicely performed.

2. The authors imply that their DL4 coated bead-based system is amenable to large scale suspension cultures and therefore superior to other cell-free systems such as plate bound DL-4 systems. This might be true but has not been shown. The obtained T cell progenitor frequency of CB-CD34+ cells cultured with the DL4 μ bead system has been reported to be similar to that using OP9-DL4 coculture platforms based on a previous publication from the same group. To show that the bead system is indeed superior to other cell free systems a side by side comparison with plate bound DL4 ligand should be performed. Ideally one should seed the same number of CD34+ CB cells in both systems and compare the T cell progenitor frequency, kinetics and phenotypes of the developing T cell progenitors.

3. CD34+CD7+ pro T cells were sorted from day 7 cultures and injected intrahepatically into neonatal NSG mice to show that DL4- μ bead derived pro-T cells can reconstitute the thymus and colonize secondary lymphoid organ as functionally mature T cells.

It would be informative to have some more insight about those functional and mature T cells. For example, was self-tolerance established in these mice? Did the authors find regulatory T cells? Showing that in vitro stimulated (anti-CD3/CD28) splenocyte derived T cells from the spleen can produce IFN- γ and TNF α , is somewhat rudimentary. Could the authors come up with more functional assays? Would it be possible to show that these T cells can mount an antigen specific CTL response?

Minor comments.

4. Numbering of the figures is messed up. For example, Fig. 3g should be 2g, and 3h should be 2h., please verify and correct.

Reviewer #1:

Many steps of T cell development can be recapitulated in vitro using stromal cells expressing Notch ligands, or by immobilizing Notch ligands on tissue culture plates. Trotman-Grant et al. refine an approach to generate mature T cells at large scale without supporting cells. Such a method will allow scalable production of T cells in a non-xenogenic stromal cell-free system that is compatible with clinical applications.

The authors use a bead-based method in which DL4-Fc is immobilized onto surface of microbeads that can activate Notch and supports differentiation of mouse and human hematopoietic stem and progenitor cells. Data shown in the paper suggest that the method of attaching DL4 to microbeads supports differentiation of hematopoietic progenitors to proT and mature T cells. In my opinion, the improvements are not clearly established, and new scientific insights are not provided in the present manuscript. It is likely that the new method is indeed scalable, and will allow some clinical applications. However, such applications are not yet demonstrated, and the work appears preliminary.

1. Cell-free systems to generate T cells have been described (PMID 29208547). It was earlier shown that using plate bound DLL4 Fc-fusion protein, adult HSPCs can efficiently generate in vitro cultured T-cell precursors that give rise to functional T-cell compartment in vivo. The authors did not compare their method with existing methods, and I am uncertain whether the bead-based method is more efficient than the plate bound method, although the beads are of course scalable. Data in Fig 1h showed bead-based method gives 400-fold expansion on d14 as compared to day0, but comparison to plate-bound methods is not performed.

We appreciate the reviewer's comments regarding the need to compare our bead-based approach to existing plate-bound methods. To this end, we are now providing a **new Figure 2** showing a direct comparison between plate-bound DL4-Fc, used at different concentrations, and DL4- μ beads. These findings show that our new approach is as efficient as the current plate-bound methods at generating T-lineage cells. Nevertheless, to address the scalable nature of this approach we have employed the Gas Permeable Rapid Expansion (G-Rex) system, which is allowed to demonstrate the scalability of the bead-based platform. We thank the reviewer for these suggestions which we feel have improved our manuscript.

2. The authors do not provide significant scientific insights. It is possible that investigation of why the highest activation of Notch signaling is seen with 25micron size bead and not smaller or larger than 25micron size may provide such insights, as the authors discuss, but these observations are at present not understood.

The reviewer brings up an excellent point regarding the bead size requirement for optimal Notch signaling. We have added to the discussion a possible rationale that helps to explain why a 25 μ m size would deliver the necessary Notch receptor engagement to trigger its activation. In essence, we feel that the necessary force to induce the pulling of the Notch receptor is

achieved by a bead that has the corresponding mass and acceleration to yield the 4-12 pN force established in the literature (PMID: 27167603).

Minor points:

1. Supplementary Fig 2 is not cited in the text.

We apologize for the oversight and we have corrected this in the revised manuscript.

2. In line 157, authors likely mean Fig2f instead of Fig2e.

We have corrected this in the revised manuscript.

3. In Fig 2f, sc-RNA seq data appears to identify a small cluster of cells that expresses LMO4 and also expresses PTPRC and HES1. But the authors state in text line 162-163 that this small cluster is not Notch-responsive and characterized by lack of PTPRC. The authors should explain the discrepancy between text and data.

The reviewer is correct in pointing out this discrepancy, which we have corrected in the revised manuscript by indicating that the small LMO4-high cluster does indeed show evidence for HES1 expression. However, the low levels of NOTCH1 and DTX1 suggest an alternative induction of HES1 expression among these cells. Nevertheless, the high levels of LMO4 expression suggests that this cluster likely represents group 2 innate lymphoid cells (ILC2) (PMID: 31434684), which we have indicated in the revised manuscript. We thank the reviewer for bringing this to our attention.

Reviewer #2:

The manuscript describes the development of a stroma-free system for inducing T lineage development from mouse and human hematopoietic stem and progenitor cells (HSPC) and human induced pluripotent stem cells (iPSC). The investigators have replaced the stroma component used in other systems by using microbeads that present DLL4 to the hematopoietic cells. They also use a commercially available serum-free medium that has been developed for T cell differentiation. The main message of the paper is that avoidance of the stroma cells will allow more simple clinical translation for adoptive cellular therapy. The data shows convincingly that the bead method induces T cell lineage commitment to the ProT stage but positive selection and maturation in vitro does not occur. The lack of data on FACS analysis, cell yield and function of iPSC derived cells limits interpretation of the technology.

Major revisions:

1. The following statement in the abstract is misleading :” Here, we report a stromal cell-free, microbead-based approach that supports the efficient in vitro development of both human progenitor T (proT) cells and mature T cells from CD34+ cells sourced from cord blood, GCSF-

mobilized peripheral blood, and pluripotent stem cells (PSCs).” No evidence is given that mature T cells (SP4 or SP8) are generated from these populations in vitro. Mature SP cells are only shown in mouse thymus and spleen after in vivo transfer into NSG mice of human proT cells generated with beads from CB.

We agree with the reviewer’s concerns that we had not fully characterized the generation of mature T cells using the DL4- μ bead system, and we have changed the emphasis of our findings, by focusing on the generation of progenitor and T-lineage cells. With this in mind, we have removed or modified statements regarding the generation of mature T cells throughout the manuscript. We feel that additional work, beyond the scope of the present manuscript, will be needed to demonstrate the effective generation of mature T cells from pluripotent stem cells.

2. The statement in the abstract “DL4- μ beads, along with lymphopoietic cytokines, induced an ordered sequence of differentiation from CD34+ cells to CD34+CD7+CD5+ proT cells to CD3+ $\alpha\beta$ T cells should also be modified.

To address this important point, the revised manuscript includes a **new Figure 4** showing the expression of CD3/TCR $\alpha\beta$ in cells differentiated using the DL4- μ beads, which was not included in the original submission.

3. T cell differentiation is shown mostly as single flow cytometry profiles with insufficient information on how the profiles have been generated. The cell populations shown in the summary graphs do not add up to 100% of the cells in culture-what are the other phenotypes? In all FACS derived data please show how the profiles shown have been gated.

The reviewer brings up an excellent point regarding the summary graphs, which represented gating profile subsets for markers that were not shown on the flow plots displayed in the figure. We have updated the figure legends to clarify this important point. We have also included a **new Supplementary Figure 2** to clarify gating strategy used for the analysis of all human T cell differentiation experiments.

4. In vivo data is shown only as representative flow cytometry plots-summarized quantitative data should be given to understand reproducibility.

We agree with the reviewer’s comment about the need to include quantitative data for the in vivo engraftment experiments, which are now provided as a **new Figure 5c**.

5. As the paper emphasizes the potential advantage of the bead system for scale-up capabilities, more detailed information on cell yield (total and each subpopulation) should be provided to get a sense of how far the process needs to be scaled to produce a therapeutic cell dose. Interesting information on the frequency of cells with CD7+ potential in different sources has been given, and fold increase in bulk cultures is shown, but it is unclear how many cells were used to initiate the cultures and how many were recovered i.e. how close are these cultures to the cell dose required for a patient? In the case of iPSC how many cells were

produced at each step of the procedure? e.g. iPSC to EBs to CD34+ to CD7+ and DP. Can the iPSC derived cells be expanded after culture?

A similar concern was raised by reviewers #1 and #3, which we have addressed by showing the scalable potential of the bead-based platform by using the G-Rex system, included as **new Figure 2**. Regarding iPSC expansion, these results are included in **Supplementary Figure 7**. We have added a statement in the discussion regarding the feasibility to reach the cell dose required for patient applications. Unfortunately, we have not addressed whether iPSC-derived T cells can be further expanded after culture.

6. Because the focus is on translation, it would be useful to know how this method compares to at least one of the existing stromal-based methods in terms of yield and differentiation, and also show some control data on cultures without beads.

A similar concern was raised by reviewer #1, which we have addressed above.

7. Functional data in the paper is limited to cells recovered from the spleen of NSG mice after infusion of proT cells from CB cultures. It is important to know how the iPSC generated cells behave functionally. How do the cells compare to those from thymus and/or peripheral blood?

We acknowledge that this was not properly addressed, which has led us refocus the manuscript to emphasize the generation of progenitor and T-lineage cells. Nevertheless, our results demonstrate the potential for in vivo use of proT cells obtained from DL4- μ bead cultures.

Minor revisions:

1. Authors show importance of bead size for notch signaling using the notch reporter cell line 3T3N1C. Are there any data on effect of bead size on T cell differentiation or yield from iPSC and/or HSPC?

This is an interesting question, however, we limited our analysis of bead size to Notch signaling outcomes.

2. Engraftment of NSG thymus by CB derived proT cells is shown as a frequency of human cells. As NSG mice contain little if any murine hematopoietic cells frequency of human cells will be high. How many human cells are recovered from each thymus?

This is an important point, which we have addressed as part of **new Figure 5c**.

3. LDA is given for "T cell Progenitor" output based on a definition of CD45+CD56-CD7+: does this define a progenitor or rather all T lineage cells?

The LDA measures T cell progenitor frequencies as a function of their ability to generate T lineage cells, defined as CD45+CD56-CD7+.

4. Legends for LDAs (Fig 1i and Fig 2e) mention p values but the figures do not seem to show *

The values for significance for the LDA results are shown in the Supplemental Tables.

5. Please specify what iPSC line is being used in each figure -e.g. Supplem Fig 7 legend

We have provided the iPSC line name in the Methods section and indicated that all PSC-related data was generated using IPS11 cells unless indicated otherwise.

6. Supplemental figures 5, 6 legends should explain the cell source for the UMAP data

This information was initially included in the Methods section, but we have now also included it in the figure legends.

7. Legend of Fig 3d should state method of stimulation.

This information was initially included in the Methods section, but we have now also included it in the figure legends (now Figure 5d).

Reviewer #3:

T cell based immuno-therapies rely largely on the expansion of autologous T cells. In vitro de novo generation of T cells from human progenitor cells for clinical applications is currently not established in part due to difficulties of producing large T cell numbers using non-xenogenic stroma cell free systems that would be compatible with clinical applications.

In this manuscript the authors report the development of a stromal-cell free, DL4 coated microbead-based system that allows in vitro development of both murine and human progenitor T cells and mature T cells.

The authors first optimized bead size for the DL4 bearing μ beads as well as bead to cell ratios for their stromal cell-free T cell differentiation system.

They used this system and tested different sources of human hematopoietic progenitor cells for their ability to efficiently differentiate into T cells, these included CD34+ cells derived from human cord blood cells, GCSF-mobilized peripheral blood and human iPSCs. The frequency with which these cells differentiated into T cell progenitors differed between the cell source with following hierarchy: CB > mPB > iPSCs . Single cell RNA sequencing was used to characterize the T cell specific gene expression pattern and adoptive transfer of CD34+CD7+ pro T cells into NSG mice was performed to demonstrate that the in vitro generated T cell progenitors were able to efficiently engraft the thymus and to functionally mature into peripheral T cells.

This is a very interesting study and to a large extend very well performed. It has the potential to be further developed so that it is suitable for publication.

Major comments:

1. The optimization and identification of the optimal bead size and bead to cell ratio is very nicely performed.

We thank the reviewer for their kind statement and support.

2. The authors imply that their DL4 coated bead-based system is amenable to large scale suspension cultures and therefore superior to other cell-free systems such as plate bound DL-4 systems. This might be true but has not been shown. The obtained T cell progenitor frequency of CB-CD34+ cells cultured with the DL4 μ bead system has been reported to be similar to that using OP9-DL4 coculture platforms based on a previous publication from the same group. To show that the bead system is indeed superior to other cell free systems a side by side comparison with plate bound DL4 ligand should be performed. Ideally one should seed the same number of CD34+ CB cells in both systems and compare the T cell progenitor frequency, kinetics and phenotypes of the developing T cell progenitors.

A similar recommendation was given by the other reviewers, we have addressed as detailed above.

3. CD34+CD7+ pro T cells were sorted from day 7 cultures and injected intrahepatically into neonatal NSG mice to show that DL4- μ bead derived pro-T cells can reconstitute the thymus and colonize secondary lymphoid organ as functionally mature T cells. It would be informative to have some more insight about those functional and mature T cells. For example, was self-tolerance established in these mice? Did the authors find regulatory T cells? Showing that in vitro stimulated (anti-CD3/CD28) splenocyte derived T cells from the spleen can produce IFN- γ and TNF α , is somewhat rudimentary. Could the authors come up with more functional assays? Would it be possible to show that these T cells can mount an antigen specific CTL response?

We appreciate the reviewer's comments, unfortunately, we limited our analysis to ex vivo activation of splenocyte derived human T cells. We agree that it would have been nice to extend our functional analysis, however, these will be done as part of future studies. Regarding whether regulatory T cells were present in NSG mice given human proT cells, this was seen in a previous publication (PMID 31648315) whereby human proT cells generated on OP9-DL4 cells were used.

Minor comments.

4. Numbering of the figures is messed up. For example, Fig. 3g should be 2g, and 3h should be 2h., please verify and correct.

We thank the reviewer for pointing this out, and apologize for the oversight, which we have corrected in the revised manuscript.

Editorial Comments:

Please provide both an accession code and hyperlink for the GEO data and a hyperlink for the ArrayExpress data and refer to these codes in your response to the reviewers and in the Data Availability section.

We have now included the GEO accession code (GSE169276) for the single-cell RNA-seq analysis data presented in the manuscript and include the link here

(<https://www.ncbi.nlm.nih.gov/geo/query/acc.cgi?acc=GSE169279>).

We have also included the accession code for the human thymus data sets obtained from ArrayExpress (E-MTAB-8581) and include the link here

(<https://www.ebi.ac.uk/arrayexpress/experiments/E-MTAB-8581/>).

REVIEWER COMMENTS

Reviewer #1 (Remarks to the Author):

This is well revised, and I am supportive.

Reviewer #2 (Remarks to the Author):

The paper has been improved in terms of data presentation and the conclusions are now more accurate with the removal of statements regarding mature T cell production. The new data comparing beads and plate-bound DLL4 show no difference in terms of differentiation and yield so the main advance over the older technology appears to be potentially easier manufacturing scale-up in a bioreactor. The rationale for the application of the bead system for cancer immunotherapy would be much stronger if the cells were engineered for antigen specificity.

Main comments:

1. Authors state (line 213) that the data shows a robust capacity for generating CD3/abTCR cells from multiple sources of stem cells. this seems to be an overstatement: no CD3/abTCR data on human HSPC is shown and it is not clear whether the CD3/abTCR generation in Figure 4b and Fig S8 is from TiPSC or clone 11 iPSC. If these data are from the STiPSC cultures these have a pre-arranged TCRA so maturation is not induced via the beads. It would therefore be more accurate to state that robust capacity for generating ProT cells is shown from multiple sources.

2. Although the methods provide some information on iPSC source, it should make clear what is the iPSC source of the data (IPS11 or STIPS) for all iPSC data shown (e.g. legends in Fig 4b and Supp Fig 8) as this affects marker expression. Also gating of these data should be explained. e.g. are the % shown based on a CD5+CD7+ gate or total live cells or CD45+ cells? it should also be clarified in the text and figure that the abTCR+ cells shown are DP and not SP. An additional panel showing CD4 v CD8 after gating on the abTCR+ cells could clarify this also.

3. Related to point 2, all legends for figures that show data on frequency of a phenotype could still be improved by describing what the denominator is for the %. Although methods and supplemental figure 2 describe this generally for CB cultures, the specifics of each figure are also needed, particularly for the iPSC cultures in Figs 3 and 4. In figure 4, are the plots shown first gated on CD5+CD7+ or CD7 only? and what are the CD5 and CD7 expression of the iPSC cultures at these later time points.

4. The presentation of the scRNA seq analysis of iPSC cultures in Fig 3 and S5 could be clearer. What stage of culture are the iPSC derived from? Are they all from iPSC11? The iPSC generate DN with little if any DP or SP-what defines a T lineage DN population and what is the frequency of other non-T cell types (particularly NK cells) that are included in this population? It is difficult to understand the info in Supplem fig 5b as the iPSC derived cells are such a small population. It would be useful to identify where they sit in the absence of fetal cells.

5. Yield is described throughout as fold increase over CD34+ input cells. For the iPSC derived cultures, what was the output of CD34+ cells per iPSC? how much more scale up would be needed for a clinical product?

6. The introduction implies that stroma and plate-bound systems are incompatible with the clinic translation but this is not yet known to be the case for all systems. The plate bound method has been used in clinical trials and xenogeneic methods while complex are not strictly prohibited. It would be more accurate to state that the bead system provides a potentially more simple/feasible system for clinical therapies than these other approaches.

Minor:

1. Fig 1 title describes only mouse data but human CB data is also included in the figure

2. Why are the LDA readout (Fig 1, 2, 3) shown as CD7+CD56- rather than the way the rest of the

data on proT cells defined (CD7+CD5+)? It is possible that non-T cells (innate like cells including NK cells) would be included in CD7+ population. Also Fig 1i legend describes these as T cell progenitors but CD34 does not seem to be included in the panel. Would T cell precursors be a better term to use if they are not CD34+?

3. Fig 1g: results state CD7+CD5+ are 80% of the culture but mean appears to be closer to 70%
4. Fig 3: What lineage potential do the authors think is included in the CD5-CD7- CD34+ cells that dominate the culture at D14? Although these are CD56- at time of analysis do they still have NK potential?
5. Fig 3b: what is center histogram showing 97.4% CD34+ gated on?
6. Fig 3f: in legend and results please state from what stage of culture are the cells that are analyzed by scRNA-Seq
7. Fig 3h and associated supplem figs: what are the exact phenotypes used for the "proT cells" isolated from the iPSC cultures and is it the same for prenatal human thymocytes?
8. Fig 5: were there any CD3a+TCRab+ in 4 week thymus? Legend does not state the n for 5c and 5d. Again-legend should explain the gating for the data presented.
9. supplementary fig 2 and 3 are not called out in text of results. Should they be reversed in order to match the text?

Reviewer #3 (Remarks to the Author):

The authors addressed most of my major concerns satisfactorily and have thereby improved the manuscript nicely. I also read the comments of the other reviewers and how the authors addressed their points. Overall, I think the manuscript is now suitable for publication.

Reviewer #1:

This is well revised, and I am supportive.

Reviewer #2:

The paper has been improved in terms of data presentation and the conclusions are now more accurate with the removal of statements regarding mature T cell production. The new data comparing beads and plate-bound DLL4 show no difference in terms of differentiation and yield so the main advance over the older technology appears to be potentially easier manufacturing scale-up in a bioreactor. The rationale for the application of the bead system for cancer immunotherapy would be much stronger if the cells were engineered for antigen specificity.

We appreciate the reviewer's vision for the applicability of our findings for potential immunotherapies, but we feel that this would necessitate its own study to fully develop these aspects of our experimental approach.

Main comments:

1. Authors state (line 213) that the data shows a robust capacity for generating CD3/abTCR cells from multiple sources of stem cells. This seems to be an overstatement: no CD3/abTCR data on human HSPC is shown and it is not clear whether the CD3/abTCR generation in Figure 4b and Fig S8 is from TiPSC or clone 11 iPSC. If these data are from the STiPSC cultures these have a pre-arranged TCRA so maturation is not induced via the beads. It would therefore be more accurate to state that robust capacity for generating ProT cells is shown from multiple sources.

The reviewer's raises a concern regarding the capacity of DL4- μ beads to generate CD3/ $\alpha\beta$ TCR from multiple sources of stem cells. Our results demonstrate a clear progression of IPSC11-derived CD34⁺ cells to about 75% CD3/ $\alpha\beta$ TCR cells by D42 (Fig. 4b). However, as the reviewer noted, Line 213 indicated a robust capacity for generate these cells from all sources of stem cells, which we had not shown. We agree with this point, and thus we have now corrected this statement in the revised manuscript.

All PSC data, unless otherwise indicated, is generated using fibroblast-derived IPS11 cells. Therefore, maturation into TCR-bearing cells is indeed induced by the DL4- μ beads.

2. Although the methods provide some information on iPSC source, it should make clear what is the iPSC source of the data (IPS11 or STIPS) for all iPSC data shown (e.g. legends in Fig 4b and Supp Fig 8) as this affects marker expression. Also gating of these data should be explained. e.g. are the % shown based on a CD5+CD7+ gate or total live cells or CD45+ cells? it should also be clarified in the text and figure that the abTCR+ cells shown are DP and not SP. An additional panel showing CD4 v CD8 after gating on the abTCR+ cells could clarify this also.

The reviewer brought up a similar point previously, which we addressed by providing a statement in the methods section indicating that, “IPS11 cell line was used to generate all PSC-related data unless indicated otherwise”. However, we appreciate that this may not be sufficient, and we now have further indicated this in the figure legends to clarify any confusion.

The % shown is based on CD45+ gated cells as indicated in Supplementary Figure 2. We have further clarified this gating strategy in the figure legend.

3. Related to point 2, all legends for figures that show data on frequency of a phenotype could still be improved by describing what the denominator is for the %. Although methods and supplemental figure 2 describe this generally for CB cultures, the specifics of each figure are also needed, particularly for the iPSC cultures in Figs 3 and 4. In figure 4, are the plots shown first gated on CD5+CD7+ or CD7 only? and what are the CD5 and CD7 expression of the iPSC cultures at these later time points.

We thank the reviewer for this suggestion. The gating strategy outlined in Supplementary Figure 2, was applied to all human T-lineage analyses, irrespective of the stem cell source. However, we agree that this would benefit by being further clarified in the text. In Figure 4, all the plots are gated on DAPI- CD45+ cells. This point has been added to the legend.

4. The presentation of the scRNA seq analysis of iPSC cultures in Fig 3 and S5 could be clearer. What stage of culture are the iPSC derived from? Are they all from iPSC11? The iPSC generate DN with little if any DP or SP-what defines a T lineage DN population and what is the frequency of other non-T cell types (particularly NK cells) that are included in this population? It is difficult to understand the info in Supplem fig 5b as the iPSC derived cells are such a small population. It would be useful to identify where they sit in the absence of fetal cells.

iPSC-proT cells were derived from IPS11-CD34⁺ cells cultured with DL4- μ beads for 7 days. This is now clarified in the legends (Fig, 3, Suppl.5) of the revised manuscript. For further clarity, we have also enhanced the color labeling of the iPSC-derived subset shown in Suppl. Figure 5b. We thank the reviewer for their comments on the frequency of alternative lineages that may arise from the PSC-derived DN population. Limiting dilution assays or bulk cultures with the endpoint being non-T-lineage CD56⁺ NK cells would have been an interesting experiment and will be explored in subsequent studies.

5. Yield is described throughout as fold increase over CD34+ input cells. For the iPSC derived cultures, what was the output of CD34+ cells per iPSC? how much more scale up would be needed for a clinical product?

The reviewer asks about the cell yield of CD34+ cells per iPSC input, which we showed as a percentage from a d8 EB culture in Figure 3b. We now provide the approximate range of cell yield per iPSC, which is derived from the change in cellularity from d0 to d8 and the percentage of CD34+ cells in those cultures. This calculation shows that 10 iPSCs give rise to 1 CD34+

postMACS-enriched cell. Regarding the need for scale up for a clinical product, this would of course depend on what the final product would be, and how much additional expansion would be needed for the generation of proT or TCR+ T cells.

6. The introduction implies that stroma and plate-bound systems are incompatible with the clinic translation but this is not yet known to be the case for all systems. The plate bound method has been used in clinical trials and xenogeneic methods while complex are not strictly prohibited. It would be more accurate to state that the bead system provides a potentially more simple/feasible system for clinical therapies than these other approaches.

We understand the reviewer's argument and agree that a more accurate statement would be to say that current systems "present challenges" to clinical translation, rather than being "incompatible". We have modified the text to reflect this sentiment.

Minor:

1. Fig 1 title describes only mouse data but human CB data is also included in the figure

We thank the reviewer for noting this and have corrected this in the revised manuscript.

2. Why are the LDA readout (Fig 1, 2, 3) shown as CD7+CD56- rather than the way the rest of the data on proT cells defined (CD7+CD5+)? It is possible that non-T cells (innate like cells including NK cells) would be included in CD7+ population.

Also Fig 1i legend describes these as T cell progenitors but CD34 does not seem to be included in the panel. Would T cell precursors be a better term to use if they are not CD34+?

We thank the reviewer for this observation and the opportunity to clarify. CD7+CD5+ proT cells define committed T-lineage cells. However, as shown in the manuscript, the kinetics by which the different stem cell sources reach that phenotype differ. We kept our LDA readout consistently CD45+CD7+CD56- because these cells are specified, lack alternative lineage potential (including NK cells) and will therefore become T cells, which was the objective of the assay.

3. Fig 1g: results state CD7+CD5+ are 80% of the culture but mean appears to be closer to 70%

We have corrected this in the revised manuscript.

4. Fig 3: What lineage potential do the authors think is included in the CD5-CD7- CD34+ cells that dominate the culture at D14? Although these are CD56- at time of analysis do they still have NK potential?

The reviewer notes that a population of CD5-CD7- CD34+ cells dominate the PSC-DL4-μbead cultures at D14. To clarify, we observe around 10% of cells possessing this phenotype at this time point. We did not directly assess the NK potential of this population, but after further differentiation we did not observe the emergence of CD56+ cells in our culture.

5. Fig 3b: what is center histogram showing 97.4% CD34+ gated on?

This histogram is gated on live, DAPI-negative cells post MACS-enrichment of CD34+ cells. We have now clarified this in the Figure legend.

6. Fig 3f: in legend and results please state from what stage of culture are the cells that are analyzed by scRNA-Seq

IPS11-CD34+ cells were cultured with DL4- μ beads for 7 days, at which point, CD34+CD7+ cells were sorted and sequenced. We have now clarified that in our Figure legend.

7. Fig 3h and associated supplem figs: what are the exact phenotypes used for the “proT cells” isolated from the iPSC cultures and is it the same for prenatal human thymocytes?

The exact phenotype of iPSC-derived “proT cells is CD45+CD34+CD7+, which does encompass the phenotype used to depict prenatal human thymocytes.

8. Fig 5: were there any CD3a+TCRab+ in 4 week thymus? Legend does not state the n for 5c and 5d. Again-legend should explain the gating for the data presented.

We did in fact observe the presence of CD3+TCR $\alpha\beta$ cells in the 4-week thymus, though the objective of that figure was to show T-lineage engraftment and the lack of alternative lineages, which we feel is achieved. However, upon further analysis of a 12-week thymus, we did look at late lineage markers and included CD3 and TCR $\alpha\beta$ in our panel, which are displayed in Fig 5c.

We apologize for the oversight and have corrected this in the revised manuscript.

9. supplementary fig 2 and 3 are not called out in text of results. Should they be reversed in order to match the text?

This oversight has been corrected and we thank the reviewer for pointing it out. Supplementary Fig. 2 and 3 are now called out in the revised manuscript accordingly.

Reviewer #3:

The authors addressed most of my major concerns satisfactorily and have thereby improved the manuscript nicely. I also read the comments of the other reviewers and how the authors addressed their points. Overall, I think the manuscript is now suitable for publication.

REVIEWERS' COMMENTS

Reviewer #2 (Remarks to the Author):

Revisions are all satisfactory with just one request for clarification:

Re the revised statement on frequency of T lineage cells (p27, line 126) "70% of CD5+CD7+ cells"
-could the authors confirm that this is what they mean? Or is it that T lineage cells (as defined as CD5+CD7+) make up 70% of the CD45+DAPI- cells (or live hematopoietic cells) in D14 cultures?

Point-by-Point Reply

Reviewer #2:

Revisions are all satisfactory with just one request for clarification:

Re the revised statement on frequency of T lineage cells (p27, line 126) “70% of CD5+CD7+ cells” -could the authors confirm that this is what they mean? Or is it that T lineage cells (as defined as CD5+CD7+) make up 70% of the CD45+DAPI- cells (or live hematopoietic cells) in D14 cultures?

We thank the reviewer for their comments. The latter interpretation is correct. This statement refers to the frequency of CD5+CD7+ cells in D14 cultures being 70% of the CD45+DAPI- cells, we have added the statement “of the live hematopoietic cells” to further clarify this point.